# *Caenorhabditis elegans* Models to Investigate the Mechanisms Underlying Tau Toxicity in Tauopathies

**DOI:** 10.3390/brainsci10110838

**Published:** 2020-11-11

**Authors:** Carmina Natale, Maria Monica Barzago, Luisa Diomede

**Affiliations:** Department of Molecular Biochemistry and Pharmacology, Istituto di Ricerche Farmacologiche Mario Negri IRCCS, Via Mario Negri 2, 20156 Milan, Italy; carmina.natale@marionegri.it (C.N.); mariamonica.barzago@marionegri.it (M.M.B.)

**Keywords:** *Caenorhabditis elegans*, tau, tauopathy, frontotemporal dementia, microtubule-associated protein tau, proteotoxicity

## Abstract

The understanding of the genetic, biochemical, and structural determinants underlying tau aggregation is pivotal in the elucidation of the pathogenic process driving tauopathies and the design of effective therapies. Relevant information on the molecular basis of human neurodegeneration in vivo can be obtained using the nematode *Caenorhabditis elegans* (*C. elegans*). To this end, two main approaches can be applied: the overexpression of genes/proteins leading to neuronal dysfunction and death, and studies in which proteins prone to misfolding are exogenously administered to induce a neurotoxic phenotype. Thanks to the easy generation of transgenic strains expressing human disease genes, *C. elegans* allows the identification of genes and/or proteins specifically associated with pathology and the specific disruptions of cellular processes involved in disease. Several transgenic strains expressing human wild-type or mutated tau have been developed and offer significant information concerning whether transgene expression regulates protein production and aggregation in soluble or insoluble form, onset of the disease, and the degenerative process. *C. elegans* is able to specifically react to the toxic assemblies of tau, thus developing a neurodegenerative phenotype that, even when exogenously administered, opens up the use of this assay to investigate in vivo the relationship between the tau sequence, its folding, and its proteotoxicity. These approaches can be employed to screen drugs and small molecules that can interact with the biogenesis and dynamics of formation of tau aggregates and to analyze their interactions with other cellular proteins.

## 1. Introduction

For many neurodegenerative diseases—including Alzheimer’s disease (AD), progressive supranuclear palsy, Pick’s disease, cortico-basal degeneration, and post-encephalitic parkinsonism—the ability of proteins to change their conformation and generate insoluble deposits in the brain of patients is believed to play a crucial role in etiopathology [1,2,3]. This is also true for frontotemporal dementia (FTD), a heterogeneous group of progressive brain disorders due to selective frontal and/or temporal lobe atrophy and frontotemporal lobar degeneration (FTLD) and associated with changes in behavior, personality and language [1]. These diseases are grouped under the common name of “tauopathies” [3] because filamentous tau inclusions, called neurofibrillary tangles [2], were identified as the major hallmark.

Tau is a natively unfolded, microtubule-binding protein that promotes tubulin polymerization and stabilizes microtubules. This protein, encoded by the microtubule-associated protein tau (*MAPT*) gene, plays a crucial role in preserving the integrity of neurons and axonal transport [3,4,5]. Six tau isoforms are present in the human brain, which differ in terms of the presence of three (R3) or four (R4) repeated sequences placed at the carboxy-terminal end. These repeated regions are encoded by exons 9–12 of the *MAPT* gene and represent functional microtubule binding domains. The ratio between the 3R and 4R isoforms is equal in normal subjects, whereas it is altered in the brain under pathological conditions [6].

The key role of tau in driving tauopathies is supported by the fact that mutations in *MAPT* cause autosomal dominant FTD and Parkinsonism linked to chromosome 17 [7]. Among the different pathogenic *MAPT* mutations described, most fall into the genetic regions encoding for tau repeats leading to changes of the R3/R4 ratio and the functional properties of the protein, thus increasing its aggregation propensity [7,8,9,10]. Substitutions or deletions of a single tau amino acid in the microtubule-binding domain modify the capability of the protein to bind to microtubules [11,12,13]. The most widespread and best characterized *MAPT* mutation is the one resulting from the substitution of the proline at position 301 with lysine (P301L) [14].

The roles of the majority of *MAPT* mutations in driving tauopathies has been demonstrated employing genetic analysis, neuropathological tests and biochemical studies on tau. However, the pathogenic effects of some other mutations remain controversial. The substitution of a single amino acid on the same *MAPT* codon may differently affect the early tau mislocalization and solubility in vivo. Increasing evidence supports the active role of tau oligomers, more than fibrillar inclusions, in causing the detrimental neurodegeneration of tau pathologies [15]. Thus, *MAPT* mutations affecting tau oligomerization can significantly modify its toxicity.

Another relevant aspect of the tauopathy progression is represented by the ability of conformational abnormal tau to spread from cell to cell in a prion-like manner [16,17]. However, the cellular mechanisms mediating this spread and their involvement in the neurodegeneration process remain to be clarified. Our limited understanding of whether the molecular identity of tau, its phosphorylation degree, and its different isoforms can actively contribute to the clinical phenotype are in part the reason for the lack of an effective therapy for tauopathies.

In an attempt to rapidly obtain relevant information in vivo, we and others have recently established novel pre-clinical models involving the use of the non-parasitic nematode *Caenorhabditis elegans* (*C. elegans*). Although distant from vertebrates, some neuronal functions and genes encoding enzymes leading to the production of various neurotransmitters—such as acetylcholine, glutamate, serotonin, dopamine, and α-aminobutyric acid—and other elements involved in synaptic transmission, are conserved in worms [18,19,20].

Numerous transgenic *C. elegans* strains have been generated to model protein misfolded diseases and the prion-like propagation of disease proteins [21]. The first line of evidence was obtained with the expression of the amyloid β peptide, a primary component of the amyloid plaques found in the brains of patients with AD [22]. Several other transgenic models [23,24] expressing proteins responsible for central or systemic amyloidosis were then designed, including worms expressing human tau [25,26,27,28,29].

The use of *C. elegans* offers numerous advantages that make it useful to rapidly analyze in vivo the genetic and molecular functions of genes related to human diseases [24]. Unlike vertebrates, the use of *C. elegans* is inexpensive and devoid of legislative restrictions and ethical issues covering the use of higher organisms, rodents included. Importantly, the generation of new animal models will also actively contribute to implementing the “three Rs” principle of Replacement, Reduction, and Refinement on the use of vertebrate animals.

This worm is easy to use and has low maintenance costs. In a laboratory setting, *C. elegans* can be maintained at a temperature between 15 and 25 °C and grown on agar plates using *E. coli* as a food source. This nematode has six pairs of chromosomes—five pairs of autosomes and one pair of sex chromosomes—and has two sexes: the hermaphrodite (XX) and the male (XO). Each individual is composed of a constant number of cells (959 somatic cells, of which 302 are neurons in the hermaphrodite) whose position is always the same [30,31]. The morphology and synaptic connections of the neurons are fully mapped [31]. These characteristics, together with the transparency of the worm’s body, allow for the visualization of tissues and the cellular localization of endogenous or transgenic proteins tagged with fluorescent dyes.

The genome of *C. elegans*, fully sequenced in 1998 [32], is composed of about 19,000 genes, of which ~60% are homologous to humans, and about 40% of the proteins encoded by this nematode are preserved in vertebrates [33,34]. Twelve of the seventeen signal transduction systems and many of the genes involved in the development and structuring of the nervous system present in *C. elegans* are conserved in humans.

*C. elegans* reproduces very efficiently. During its life, a single hermaphrodite generates more than 300 individuals who are clones of itself, thus offering an efficient system for the generation of transgenic strains. In laboratory conditions, the life span of the worms ranges from 12 to 18 days. Thanks to this short life expectancy, this nematode is suitable for studies regarding aging and age-related diseases. A number of disorders related to senescence, including protein misfolding diseases, have been successfully modeled in *C. elegans*, usually by generating transgenic strains. This nematode also retains many of the stress response systems found in mammals, making it a fast and versatile system for recapitulating the molecular mechanisms underlying complex diseases. It has been observed that at least 42% of the genes responsible for disease in humans have counterparts in worms, suggesting that most of the biochemical pathways are conserved through evolution [35].

Transgenic *C. elegans* strains can be generated by expressing the transgene, sometimes tagged with a fluorescent protein, under a promoter that allows for tissue-specific expression in muscle, intestinal, or neuronal cells. These strains were employed to decipher the mechanisms of proteotoxicity and discover new genes related to the disease [36,37,38,39]. Microarray experiments, bioinformatic analysis, and RNA interference can facilitate understanding of the cellular and tissue responses adopted by the nematode to respond to protein misfolding-induced stress. Several studies have demonstrated that the genes controlling lifespan are the same ones involved in protein homeostasis, thereby providing a link between aging and proteotoxicity and offering a plausible explanation for the age-dependent onset of neurodegenerative disorders [40,41,42,43].

Last but not least, *C. elegans* can efficiently be used for the screening of drugs and small molecules able to interact with the biogenesis and dynamics of the formation of protein aggregates and interaction with other cellular proteins, such as chaperones [23,44,45].

We have recently established novel pre-clinical *C. elegans* models for the elucidation of the causal role of abnormal tau conformers in driving toxicity in tauopathies. In particular, we generated new transgenic strains expressing human tau in the wild-type (WT) or mutated form whose sequence was deduced from patients suffering from FTLD with different atypical clinical phenotypes and carrying mutations on codon 363 of the *MAPT* gene. We also conceived the use of *C. elegans* to recognize the forms of tau relevant for proteotoxicity.

## 2. Transgenic *C. elegans* Models of Tauopathy

Various transgenic *C. elegans* strains expressing human WT or mutated tau have been generated in the past few decades as new models of tauopathy [25,26,27,28,46,47,48]. Although *C. elegans* has a tau homolog protein with tau-like repeats (PTL-1) and a high level of sequence homology with the repeat region of mammalian microtubule-associated protein (MAP)2, MAP4, and tau [49,50,51,52], these transgenic strains are useful for deciphering the role of hyperphosphorylation and conformational changes in human tau within the context of neurodegeneration. In fact, PTL-1 is expressed only in a small subset of neurons, and its loss or mutation does not recapitulate tau pathology [49,50,51,52].

The motility of *C. elegans* is coordinated by a precise interaction between sensory and motor neurons and body wall muscle cells. Different promoters were employed to express the human tau WT at the pan-neuronal level, and the transgenic worms exhibited progressive uncoordinated movement/locomotion accompanied by accumulation of insoluble tau [25,26,28] (Table 1). A similar dysfunction was observed when other amyloidogenic proteins—such as amyloid β, α-synuclein, or huntingtin—were expressed in all the worm’s neurons [53,54]. When human tau was expressed in six mechanosensory neurons—a subset of neurons governing the touch response of worms—a decrease in the touch response, accompanied by neuritic abnormalities and microtubular loss, was observed [27] (Table 1). Worms expressing mutant human tau associated with cases of FTD and Parkinsonism linked to chromosome 17 (P301L, V337M, R406W, F3ΔK280, and A152T), as well as hyperphosphorylated tau, had more severe phenotypes and a greater accumulation of insoluble tau than *C. elegans* expressing tau WT [25,26,27,28,55]. More recently, Butler et al. [29] reported that the phosphorylation of threonine 152 in transgenic worms expressing A152T tau exerts a relevant role in neuronal toxicity via impaired axonal trafficking [29]. On the one hand, these findings helped to elucidate the mechanism underlying the ability of mutated tau to mediate the neurodegeneration in vivo; on the other, they confirmed the complexity of the mechanisms that regulate tau proteotoxicity.

We have recently generated two additional strains expressing at the pan-neuronal level human 2N4R tau carrying a V363A or V363I substitution [56] (Table 1). These mutations, identified in patients with FTLD presenting different atypical clinical phenotypes, are caused by two mutations on the same codon of the *MAPT* gene [57]. V363I substitution was previously described in asymptomatic subjects, thus raising uncertainties about its relevance in pathogenesis [58,59,60]. Studies in vitro have indicated that the two substitutions differently affected microtubule polymerization and, unlike other *MAPT* mutations, resulted in the production of tau with a high propensity to form oligomers [57]. These two new *C. elegans* strains were employed to elucidate whether the expression of the mutant genes resulted in gene-specific disease hallmarks. To this end, their functional and biochemical characteristics were compared to those of transgenic nematodes expressing human 2N4R tau WT. We found that the V363A and V363I mutations differently affected in vivo, in transgenic worms, the expression of tau at neuronal level, the amount of protein produced, and its phosphorylation degree. Like other transgenic *C. elegans* strains expressing human tau whose sequence was deduced from patients with FTD and Parkinsonism linked to chromosome 17, tau carrying a V363A or V363I mutation caused a motility defect and a comparable shortening of the worm’s lifespan [26,46]. In addition, in V363A nematodes, we observed a peculiar pharyngeal dysfunction as a consequence of an effect on neurons grouped in the nerve ring around the pharynx. This outcome had never been previously observed with other mutations, and it was later also reported in A152T and A152E strains [29]. V363A and V363I tau differently affected the synaptic transmission: V363A caused a presynaptic defect involving both motor and pharyngeal neurons, whereas V363I induced only a postsynaptic defect that resulted in motor neuronal dysfunction.

As observed in vitro, the two mutations differently influenced the ability of tau to misfold in vivo and form soluble oligomers. In particular, V363A substitution promotes the formation of more dimeric/trimeric tau assemblies and fewer monomers than V363I. Given that V363A mutation is the most toxic and phosphorylated, this finding points to the key role of soluble tau oligomers in driving the proteotoxic process involved in tauopathy.

## 3. Other Non-Mammalian Models of Tauopathy

Transgenic mice models of tauopathy have traditionally been used and preferred to non-mammalian ones due to their close evolutionary relationship with humans but also due to preconceived opinions. Because of the continuous rise in ethical concerns and legislation restrictions on the use of vertebrates, together with the need to reduce research costs, an increasing number of laboratories have begun to employ alternative models. In addition to *C. elegans*, the most commonly used are *Drosophila melanogaster* (fruit fly) and the small vertebrate *Danio renio* (zebrafish) which have also been employed to generate transgenic models of tauopathy. Like all models developed for biological and biomedical research, the use of each of these animals has intrinsic advantages and disadvantages which must be carefully examined before planning experiments, considering the question to be answered.

Like *C. elegans*, Drosophila offers the advantage of being quite inexpensive and easy to use, with a short life cycle and a high reproduction rate. Adult animals have heart, lungs, kidneys, intestine and reproductive tract functionally close to those of mammals. Its genome is fully annotated and 77% of genes involved in human diseases have homologous genes in Drosophila [61,62]. Transgenic strains of fruit flies can be easily generated as models of human diseases but, differently from *C. elegans* which can be frozen and stored indefinitely, the freezing and revival of mutant Drosophila strains is difficult.

Drosophila has a single gene homologous to a microtubule-binding domain motif in the human (*tau*) driving the expression of the protein in the developing brain, ventral nerve cord and peripheral nervous system. In adults, tau is mainly expressed in the neuronal cells of the retina and brain [63].

Some models of tauopathy were generated in fruit flies by expressing in the eye the human 4R isoform of WT tau, R406W- or V337M-muted tau [64,65]. A comparable lifespan reduction and age-dependent neurodegeneration, characterized by nuclear fragmentation and vacuole formation in neurons, was observed in all strains [65,66]. Neuronal dysfunction and defects in synaptic transmission, microtubular assembly and axonal transport were also described [67,68]. The expression of WT tau in both neuronal and non-neuronal postmitotic retinal cells of Drosophila resulted in the onset of phenotypes specific for retinal toxicity [64,66]. Trying to reproduce the memory and olfactory dysfunctions occurring in patients affected by tauopathies, bovine or human WT tau was expressed in the mushroom body neurons of flies, a center for olfactory learning and memory, and an impairment in associative olfactory learning and memory prior to the onset of neurodegeneration was observed [69,70]. Furthermore, fly models were employed to study the role of phosphorylation in toxicity. A genetic screen performed in a Drosophila expressing human WT tau identified kinases and phosphatases as important modifiers of tau toxicity in vivo, revealing the key role of glycogen synthase kinase-3β (GSK-3β) and cyclin-dependent protein kinase 5 [64,71]. To directly test the role of tau phosphorylation in mediating toxicity, the phenotype caused by the expression in fly eyes of human R406W-mutated tau was compared to those obtained when the transgene was manipulated to produce phosphorylation-incompetent or phosphorylation-mimicking Ser/Thr kinase sites tau, showing that the absence of phosphorylation reduced the neurotoxic phenotype [66,72].

In Drosophila models of tauopathies, similarly to *C. elegans* models, no neurofibrillary tangles were observed. If, on the one hand, this may represent a limitation of the models, on the other hand it could validate the hypothesis that the soluble forms and not the fibrillar ones are responsible for toxicity. Compared to worms, experiments exploring the transmission and spread of tau are more difficult to conduct in the fly and so far, there are no studies on this topic.

Zebrafish is considered as an ideal model to study human neurological disorders due to the conservation of basic brain organization [73], and of many key neuroanatomical and neurochemical pathways relevant to human diseases [74,75,76,77]. Unlike the worm, this fish has a central nervous system with microglia, astrocytes, oligo-dendrocytes and myelin, and has a blood–brain barrier [78,79,80,81,82]. It expresses neurotransmitters like those produced in the human brain, including acetylcholine, dopamine, and glutamate [83]. This fish, easily kept in the laboratory where it breeds all year, has a higher cost of maintenance compared to worms and flies [84]. Particularly diffused is the use of zebrafish embryos, which are considered as invertebrates and are not subjected to ethical and legislative restrictions. In addition, embryos have a transparent body and develop externally from the mothers, offering direct access for manipulation and observation of organogenesis. However, it should be considered that the use of embryos may represent a limit when diseases that arise in adulthood and related to aging must be modelled.

The zebrafish genome contains highly conserved orthologues of human genes, and 84% of them, including kinases implicated in tau phosphorylation, are associated with human diseases [85,86,87,88,89,90,91,92,93,94,95]. Two paralogues of the *MAPT* human gene, called *mapta* and *maptb,* were identified in zebrafish but the expression profile of the two encoded proteins has not yet been clarified. However, it is known that both *mapta* and *maptb* mRNAs are expressed in the developing central nervous system [96].

Different models of tauopathy have been generated in the last two decades. The expression in zebrafish’s neurons of human P301L tau fused with green fluorescent protein (tau::GFP) resulted in cytoskeletal disruption and accumulation of fluorescent fibrillar structures resembling neurofibrillary tangles in the cell body and proximal axon [81]. Biochemical analysis showed that the tau::GFP accumulated in the brain of both embryo and adult animals was phosphorylated and this was associated with an increased expression of GSK-3, confirming that there is sufficient phylogenetic conservation of endogenous zebrafish kinases to modify human protein. Similar results were obtained in zebrafish embryos expressing the human 2N/4R P301L tau in which an accumulation of abnormal phosphorylated tau conformers was observed. In this model, the use of GSK-3β inhibitors designed to target the human enzyme reduced phosphorylation, confirming once again the homology between human and zebrafish kinases [97].

A transgenic zebrafish strain was developed by expressing human 0N/4R tau WT in the central nervous system [98] resulting in an abundant expression of protein mainly in axons and neuropil, which persisted in adulthood. Ectopic neuronal somatic tau accumulations resembling neurofibrillary tangles seen in human tauopathies were observed. In another study, the effect of the expression of human tau WT and A152T-mutated tau was compared. In zebrafish expressing the mutant protein, a greater neurodegenerative phenotype was observed compared to those expressing the WT tau. In addition, the mutated protein caused an impairment of proteasome function and a delay in the tau clearance in vivo, which are reverted by both the pharmacological and genetic upregulation of autophagy [99].

The main concern in the use of zebrafish to model neurodegenerative diseases, including tauopathies, regards the fact that this fish has an excellent capacity for regeneration, and this may impact on the development of the neurodegenerative phenotype [84].

## 4. *C. elegans* Recognizes the Toxic Component of Tau

To study the relationship between the toxicity and structure of amyloidogenic proteins, as well as the pathogenetic mechanisms responsible for the formation of tissue deposits, different experimental approaches can be applied. These mainly involve in vitro studies aimed at recognizing and characterizing conformational states and evaluating the toxicity of different protein assemblies [100]. Particular attention is paid to the formation of soluble oligomers, very reactive intermediates formed during the protein aggregation process, which are more toxic than the end-stage fibrillar species [100].

Some in vivo studies have been performed by microinjecting into zebrafish soluble or aggregated proteins, such as human amyloidogenic immunoglobulin light chain (LC) and α-synuclein, associated with LC amyloidosis (AL) and Parkinson’s disease, respectively [101,102]. This approach, although useful, requires a high level of expertise and is not feasible for future high-throughput drug discovery purposes. Information on the toxic effects of specific protein assemblies, especially transient soluble intermediates, is difficult if not impossible to obtain in rodents, underlying the need to develop new animal models. To this end, our group has developed a totally innovative method in which *C. elegans* can be used as a “biosensor” able to react quickly and specifically to the toxic forms of amyloidogenic proteins by developing specific behavioral dysfunctions.

This method involves the direct administration to ancestral N2 nematodes of an amyloidogenic protein obtained by recombinant production or purified from the biological fluids of diseased patients, such as blood and urine (Figure 1). Alterations in *C. elegans* behavior are then assayed to monitor the onset of proteotoxic effect. The biophysical and biochemical characterization of the protein is fundamental in linking the toxicity with a specific conformational state (Figure 1).

With this approach, we demonstrated that some proteins responsible for amyloidogenic diseases caused the specific inhibition of the worms’ pharyngeal function [103,104,105,106,107]. *C. elegans* specifically recognized amyloid β oligomers as toxicants, but not monomers or larger aggregates, which are involved in the pathogenetic process of AD [108]. A similar effect was observed with human immunodeficiency virus type-1 (HIV-1) matrix protein p17 (p17), which is able to misfold and generate toxic assemblies in the brains of patients suffering from HIV-1 associated with neurocognitive disorders [104]. Our findings indicate for the first time that the ability of p17 to form soluble toxic assemblies may be a potential cause of neurodegeneration in HIV-1-seropositive patients. The *C. elegans*-based assay was also applied to investigate the mechanisms underlying gelsolin amyloidosis, a neglected familial amyloidosis caused by pathological aggregation and peripheral deposition of proteolytic fragments of plasma gelsolin [107].

The process driving LC toxicity in AL amyloidosis, the most common systemic form in Western countries, was also studied. Regardless of the tissue in which fibrillar LCs is deposited, the main prognostic determinant of this disease is cardiac involvement which leads to death from congestive heart failure or fatal arrhythmias in more than 80% of patients [109]. The elucidation of the mechanisms guiding LC toxicity uncovers an urgent need to design effective pharmacological therapies. We demonstrated that the administration to worms of LC purified from AL patients with compromised cardiac function selectively and permanently impaired the *C. elegans* pharyngeal function [105]. The biological relevance of this observation is reinforced by the fact that the worm’s pharynx is an “ancestral heart” which shares function, electrobiology and embryonic development with vertebrate heart. Biochemical, structural and molecular studies have indicated that the pharyngeal toxicity of cardiac LC is due to its intrinsic ability to interact with metal ions and generate reactive oxygen species, thus causing structural mitochondrial damage similar to that observed in the amyloid-affected hearts of AL patients [106].

We also demonstrated that the pharyngeal dysfunction caused by the different amyloidogenic proteins is specifically associated with their ability to form soluble oligomeric assemblies whose peculiar toxic effect was also confirmed in other cellular and animal models [103,104,108].

The exogenous administration of amyloidogenic proteins to *C. elegans* has also been recently applied to elucidate the possible relationship between the conformational state of tau and its biological effects [110]. First, soluble or insoluble aggregated forms of recombinant WT or P301L-mutated protein, whose conformational state was fully characterized using different biophysical and biochemical approaches, were employed. Interestingly, worms treated with oligomeric tau but not monomeric protein did not develop a pharyngeal defect but exhibited a neuromuscular dysfunction that appeared 48 h after the administration and worsened over time. The reason why tau does not act as pharyngeal stressor and results in a peculiar motoneuronal toxic phenotype remains to be clarified. The roles of ingestion, absorption and/or digestion also remain to be explained.

WT and P301L tau caused a comparable dose- and time-dependent toxic effect, suggesting that the conformational state of tau, more than sequence, is relevant to the onset of motility deficit in worms. Using aldicarb and levamisole, two pharmacological compounds employed to establish the involvement of synaptic dysfunction in neuromuscular defects [111], we also demonstrated that oligomeric tau impaired the synaptic transmission in *C. elegans*, affecting both pre- and post-synaptic cholinergic neuronal function. These defects are comparable to those scored in transgenic worms expressing WT or mutated tau [26,56], suggesting that the toxicity observed using the two models may involve similar mechanisms.

To better validate our observations, we conceived the use of brain homogenates from nine-month-old P301L transgenic mice, which express high levels of human hyperphosphorylated tau and represent a well-characterized animal model of tauopathy [14]. Brains from non-transgenic mice of the same age were employed as negative controls. We observed that the administration to worms of brain homogenates from P301L mice, but not non-transgenic ones, caused a neuromuscular defect comparable to that observed with oligomeric recombinant tau. The connection between misfolded/oligomeric tau and neuronal impairment was supported by the fact that P301L homogenates were no longer toxic when incubated with anti-tau antibodies [112]. These results indicate that the toxic form of tau that accumulates over time in an animal model of tauopathy damages the *C. elegans’* neuromuscular control, indicating a crucial role of abnormal tau conformers in chronic neurodegeneration. This worm-based approach can be employed to gain insights into the complex molecular interactions that drive neurological dysfunction and may represent an experimental platform to screen for pharmacological agents.

## 5. *C. elegans* for Drug Discovery

The ultimate goal of the use of *C. elegans* for the elucidation of the relationship between protein misfolding and a cascade of dysfunctions involved in amyloidogenic diseases is to identify new molecular targets useful for the develop of effective therapeutic approaches. However, it should be remembered that *C. elegans* is a complementary and not an alternative approach to the use of vertebrate animals and, although highly informative, the evidence obtained in worms needs to be validated in mammalian systems.

The generation of a transgenic strain modelling a human disease and with specific phenotypes is one of the strategies that can be applied for target identification. Genetic approaches can then be used to validate the relevance of the identified target genes. These involve the silencing or mutation of the gene through the use of genome-wide RNAi libraries or classical mutagenesis, and the subsequent identification of worms in which the diseased phenotype is reverted [113].

The ability of numerous compounds acting on the identified targets thus reverting or attenuating the toxic phenotype have been tested in transgenic worms developed as models of neurodegenerative disorders, such as Alzheimer’, Parkinsons’ and Huntington’ diseases [113]. Transgenic *C. elegans* models of tauopathies have also been recently employed to search for effective anti-tau therapeutic strategies, although their use for this purpose is less diffused compared to the other amyloidogenic models. Employing worms expressing WT or R406W-mutated tau at pan-neuronal level, Miyasaka et al. evaluated the protective effect of curcumin, methylene blue, resveratrol and trehalose [55]. Among them, curcumin resulted in being the compound most effective in reverting the behavioral abnormalities in tau-expressing worms. Its effect is not linked to a reduction of tau expression, phosphorylation or formation of insoluble aggregates, but seems specifically due to curcumin’s ability to protect neurons from the tau-induced dysfunction.

The relationship between abnormal glucose level and tau hyperphosphorylation was investigated pointing to the role of dihydrolipoamide dehydrogenase gene (*dld*), encoding for mitochondrial enzymes with a vital role in energy metabolism. This study stems from previous observations indicating that the manipulation of this gene through its direct suppression by RNAi, or the inhibition of the dihydrolipoamide dehydrogenase enzyme activity (DLD), protects transgenic worms expressing human amyloid β from protein toxicity [114,115]. Transgenic *C. elegans* expressing in neurons the human fetal tau, formed by 352 amino acids and three microtubule-binding domains, were employed [116]. Although this strain, characterized by a defect in the synaptic transmission of cholinergic neurons, does not represent a model of tauopathy, it can be employed to investigate the role of tau hyperphosphorylation in toxicity [117]. Inhibition of DLD by 5-methoxyindole-2-carboxylic acid, as well as the suppression of the *dld* gene, increased worms’ glucose levels, induced tau phosphorylation and reverted the neurotransmission defect. These findings indicate that an impairment of energy metabolism can affect the post-translational modification of tau, opening new ideas for pharmacological relevant targets.

The effect of NP103, a highly specific inhibitor of the main kinase involved in tau phosphorylation, has recently been tested in transgenic *C. elegans* expressing V337M-mutated tau, and proved effective in improving the motility defects of worms [118]. In the same study, Gamir-Morralla et al. also evaluated the protective effect of thioflavin T (ThT), a known anti-aggregating compound structurally related to tetracyclines. These antibiotics, thanks to their peculiar pleiotropic action, proved effective in a variety of pre-clinical and clinical studies involving amyloidogenic diseases [119]. ThT protected the transgenic worms from the V337M tau-induced locomotor dysfunction and shortened lifespan, suggesting the consideration of tetracycline use for future experiments.

The use of *C. elegans* as biosensor able to recognize protein toxicity represents an additional approach to identifying druggable targets and testing new treatments. This method has not so far been used to test the effect of anti-tau compounds, but we intend to apply it soon. It was, however, employed to assess the activity of a variety of molecules on the in vivo toxicity of Aβ oligomers and to elucidate their mechanism of action. Clusterin, a molecular chaperone, and epigallocatechin gallate (EGCG), a green tea-derived polyphenol with known antioxidant activity, were found effective in protecting worms from Aβ oligomer toxicity and able to prevent their formation [120]. We also demonstrated that small molecules, such as the Aβ1-6 peptide carrying the alanine-to-valine substitution and the N1 fragments of cellular prion protein, bind to oligomers during Aβ polymerization and counteract their toxicity [103]. These observations were further substantiated by the ability of N1 to protect cultured hippocampal neurons against the Aβ oligomer toxicity in vitro and to prevent mice from the memory dysfunction caused by intracerebroventricular injection of synthetic Aβ oligomers [103].

The key role of metal ions in driving the oxidative damage of LCs responsible for AL amyloidosis and cardiac impairment was revealed for the first time thanks to the use of *C. elegans* as biosensor. Antioxidant drugs and tetracyclines, able to act as antioxidant compound and mild metal chelator, protected worms from pharyngeal damage [105]. We also conceived that metal-binding 8-hydroxyquinoline compounds clioquinol and PBT2 can definitively block LC-induced redox damage and demonstrated that their combined use with tetracyclines resulted in a synergic protective effect [106]. In the absence of animal models for AL amyloidosis and due to the need to improve the survival of patients, the information obtained in *C. elegans* has been for the first time directly translated into clinical application supporting the design of clinical studies (ClinicalTrials.gov Identifier: NCT03474458 and NCT02207556) [109].

## 6. Conclusions

Despite the efforts of numerous groups in recent decades, the molecular mechanisms underlying tauopathies remain unknown, and an effective pharmacological therapy is still lacking. The nematode *C. elegans,* being less complex but easier to manage than vertebrates, represents an effective informative system for rapidly verifying new hypotheses. Using strains trans-genically expressing mutant tau in the absence of homologs, it represents an ideal system to investigate the cell-autonomous mechanisms underpinning chronic neurodegeneration as well as the non-cell-autonomous mechanisms underlying the spreading of tau (Figure 2). Our recent observation of the ability of *C. elegans* to be sensitive specifically to the toxic component of tau, even when it is administered exogenously, opens up the use of this assay to investigate in vivo the relationship between the tau sequence, its folding, and its proteotoxicity (Figure 2).

This nematode-based approach can be employed as an experimental platform to screen for pharmacological agents that interfere with pathologies associated with tau toxicity (Figure 2). In addition, *C. elegans* offers unprecedented opportunities to gain insight into the possible role of the interplay between tau and other amyloidogenic proteins in driving the neuropathogenic process of tauopathies.

## Figures and Tables

**Figure 1 brainsci-10-00838-f001:**
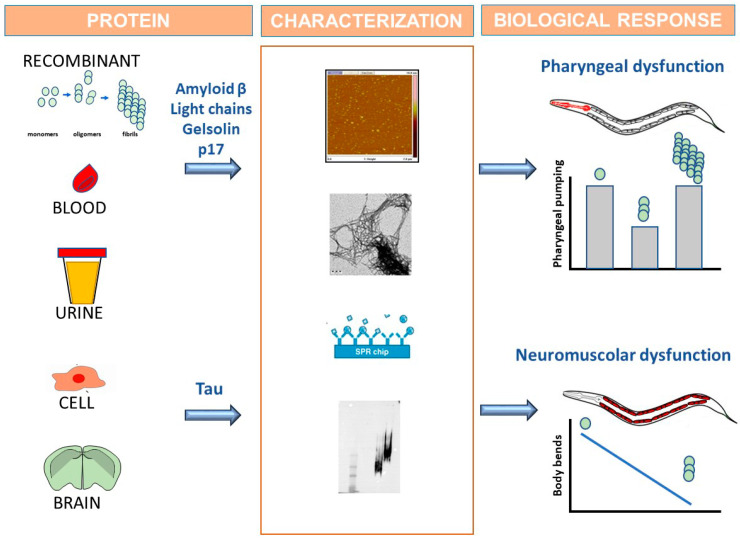
*C. elegans* use as biosensor able to recognize the toxic forms of amyloidogenic proteins by developing specific behavioral dysfunctions. Amyloidogenic proteins are obtained by recombinant production or purified from the tissues of transgenic animals or diseased patients. Homogenates from cells expressing human amyloidogenic proteins or brains from animals modelling amyloidosis can also be employed as protein source. The conformational state of the proteins must be characterized by employing biophysical and biochemical methods (i.e., atomic force microscopy, electron microscopy, surface plasmon resonance, native and denaturing gels). Proteins are administered to worms and alterations in the *C. elegans* behavior are then assayed to monitor the onset of proteotoxicity in vivo. Data obtained with synthetic amyloid β peptides, recombinant gelsolin, human immunodeficiency virus type-1 matrix protein p17 (p17) and amyloidogenic light chains purified from the blood or urine of patients with light chain amyloidosis, showed that the soluble oligomeric assemblies of these proteins, but not monomers and fibrils, caused a pharyngeal dysfunction in worms. The administration to nematodes of recombinant wild-type tau or tau in which proline at position 301 is substituted with lysine, as well as mutated protein from brain homogenates of transgenic mice, causes a neuromuscular dysfunction.

**Figure 2 brainsci-10-00838-f002:**
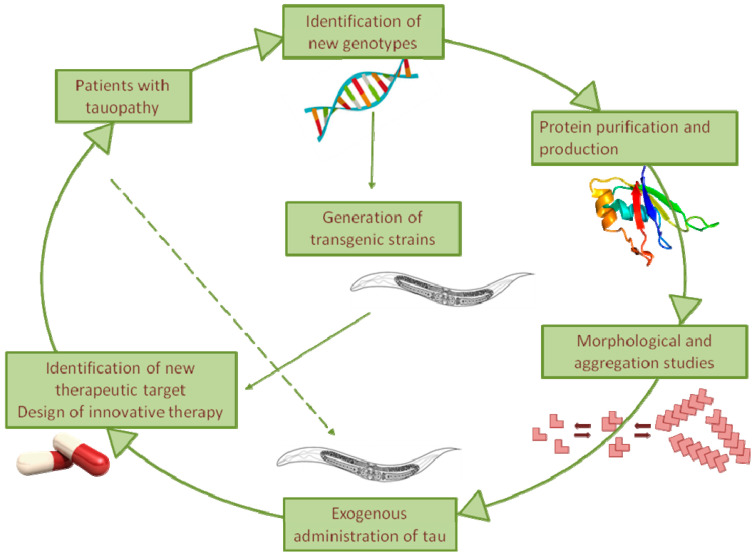
Overview of the *C. elegans* use as model for the elucidation of the pathogenic processes driving tau toxicity in tauopathies as well as the design of effective therapies.

**Table 1 brainsci-10-00838-t001:** Transgenic *C. elegans* models of tauopathies related to microtubule-associated protein tau (*MAPT)* gene mutation.

Promoter::Transgene	Expression Pattern	Phenotype	Reference
*Paex-3::tau* WT *Paex-3::tau* V337M *Paex-3::tau* P301L	Pan-neuronal	Uncoordinated movement Nerve cord degeneration Insoluble tau accumulation	[26]
*Pmec-7::tau* WT *Pmec-7::tau* P301L *Pmec-7::tau* R406W	Mechanosensory neurons	Decrease in touch response Neuritic abnormalities and microtubular loss Tau accumulation	[27]
*Prgef-1::tauWT* *Prfef-1::tau PHP °*	Pan-neuronal	Uncoordinated locomotion Defect in motor neuronal development	[25]
*Prab-3::tau* F3ΔK280 *Prab-3::tau*F3ΔK280-PP *	Pan-neuronal	Locomotion impairment Motor neuron damage Tau aggregation	[46]
*Punc-119::tau* WT *Punc-119::tau* R406W	Pan-neuronal	Uncoordinated movement Neuritic abnormalities and microtubular loss	[55]
*Psnb-1::tauWT* *Psnb-1::tauA152T*	Pan-neuronal	Locomotion impairment Paralysis Neuronal dysfunction	[28]
*Paex-3::tauWT* *Paex-3::tauV363A* *Paex-3::tauV363I*	Pan-neuronal	Locomotion impairment Pharyngeal dysfunction Insoluble tau accumulation Synaptic impairment	[56]
*Paex-3::tauWT* *Paex-3::tauA152T* *Paex-3::tauA152E*	Pan-neuronal	Developmental toxicity Impaired retrograde axonal transport	[29]

° PHP tau = Hyperphosphorylated tau. Codons for S198, S199, S202, T231, S235, S396, S404, S409, S413 were changed to glutamate.* F3ΔK280-PP is and anti-aggregating strain due to the I277P and I308P substitutions.

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
