# Peer review of "Caenorhabditis elegans Models to Investigate the Mechanisms Underlying Tau Toxicity in Tauopathies"

_brainsci, 2020, doi:10.3390/brainsci10110838_

Round 1
Reviewer 1 Report
This piece of work from Natale and coworkers is a nice , well performed and organized , review, about the using of C.elegans as a model to investigate the mechanisms of underlying tau toxicity in tautopathies. It is complete and well referenced.
The introduction is well suited. The conclusions support what is explained in the text. Each of the chapters of the review is well organized and the content is adequate. However I can miss a couple of things. One thing I can miss is a chapter devoted to talking more in detail about drug discovery. How the commented C.elegans models can help, having examples of that etc. I said that because it is mentioned in the abstract that "These approaches can be employed to screen drugs and small molecules that can interact with the biogenesis and dynamics of formation of tau aggregates and to analyze their interactions with other cellular proteins" but then there are only a couple of lines inside the text. However it is a personal consideration and if the aim of the authors is not to talk about it, is ok.
In the same way it would be fine to have a chapter with a comparison between C.elegans and other models to study tauopathies. This is partially commented in text but it can be useful to have a dedicated chapter and a larger discussion. However I also think that it is a decision of the authors if they want to include it or not. I think it is not something mandatory to accept or reject the paper.
It can be improved if the authors include the two chapters I comment. Probably the most important of the two , giving the aim of the review, is the one comparing different tautopathy models.
Author Response
According to the the reviewer's request, we have added the following two new chapters:
Chapter 3, entitled “Other Non-Mammalian Models of Tauopathy” in which we discussed the use of Drosophila melanogaster (fruit fly) and the small vertebrate Danio renio (zebrafish) to generate transgenic models of tauopathy.
Chapter 5, entitled ” C. elegans for Drug Discovery” in which we talk more in detail about the employment of the use of worm for drug discovery.
Reviewer 2 Report
The review manuscript entitled "Caenorhabditis elegans models to investigate the mechanisms underlying tau toxicity in tauopathies" is a concise and very well written overwiew of C. elegans models for devastating human disorders, characterized by the accumulation and aggregation of aberrant tau variants. It is of high interest for the wide readership of the journal, and the authors can address some ammendments.
Major comments:
1.) The review would highly benefit from an additional figure illustrating key points of chapter 2 "Transgenic C. elegans models of tauopathies".
2.) Chapter 3 "C. elegans recognizes the toxic component of tau" should be revised for a better understanding. Whereas the contents are fine in this chapter, I needed to read this chapter three times to get the key points.
3.) I recommend including a brief chapter (e.g. chapter 4), in which the C. elegans models are systematically compared to other models of tauopathies, highlighting the advantages and disadvantages using this model. This kind of discussion is partially already included in the text, but I think it would be better to summarize it in one concise chapter.
Author Response
1. The review would highly benefit from an additional figure illustrating key points of chapter 2 "Transgenic C. elegans models of tauopathies".
We thank you very much the reviewer for this suggestion but is very difficult for us to conceive a new figure illustrating the key points of chapter 2.
2. Chapter 3 "C. elegans recognizes the toxic component of tau" should be revised for a better understanding. Whereas the contents are fine in this chapter, I needed to read this chapter three times to get the key points.
As required, we revised the chapter (Chapter 4 in the revised version) trying to render it more easy to read and understand. We added also a new Figure (Figure 1) illustrating the use of C. elegans as biosensor able to recognize the toxic forms of amyloidogenic proteins by developing specific behavioral dysfunctions.
3. I recommend including a brief chapter (e.g. chapter 4), in which the C. elegans models are systematically compared to other models of tauopathies, highlighting the advantages and disadvantages using this model. This kind of discussion is partially already included in the text, but I think it would be better to summarize it in one concise chapter.
According to the the reviewer request, we have add a new chapter ( Chapter 3), entitled “Other Non-Mammalian Models of Tauopathy” in which we discussed the use of Drosophila melanogaster (fruit fly) and the small vertebrate Danio renio (zebrafish) to generate transgenic models of tauopathy and in which the pros and cons of each model was considered.